# Pseudo Knowledge Distillation: Towards Learning Optimal Instance-specific Label Smoothing Regularization

## Abstract

Knowledge Distillation (KD) is an algorithm that transfers the knowledge of a trained, typically larger, neural network into another model under training. Although a complete understanding of KD is elusive, a growing body of work has shown that the success of both KD and label smoothing comes from a similar regularization effect of soft targets. In this work, we propose an instance-specific label smoothing technique, Pseudo-KD, which is efficiently learnt from the data. We devise a two-stage optimization problem that leads to a deterministic and interpretable solution for the optimal label smoothing. We show that Pseudo-KD can be equivalent to an efficient variant of self-distillation techniques, without the need to store the parameters or the output of a trained model. Finally, we conduct experiments on multiple image classification (CIFAR-10 and CIFAR-100) and natural language understanding datasets (the GLUE benchmark) across various neural network architectures and demonstrate that our method is competitive against strong baselines.

## 1 Introduction

Neural networks, predominantly, pre-trained language models have grown massively in scale over the past few years (Fedus et al., 2021; Brown et al., 2020; Zeng et al., 2021). Some researchers have gone so far as to term these over-parameterized models as *Foundation Models* (Bommasani et al., 2021). Although larger models trained on more data tend to dominate various benchmarks (Wang et al., 2019b;a; Deng et al., 2009), deploying them on edge devices is a challenge. Training smaller networks is an option but empirical evidence suggests that training larger models and then compressing them can help achieve a higher accuracy per unit computation (Li et al.).

Knowledge distillation (KD) (Hinton et al., 2015; Buciluǎ et al., 2006) which is a knowledge transfer technique, from a single or an ensemble of neural networks to another, is a widely applied neural model compression technique across domains ranging from computer vision (He et al., 2019; Xu et al., 2020; Jafari et al., 2021) to natural language processing (Jiao et al., 2020; Sanh et al., 2019). This empirical success is attributed to a few factors such as a) deeper teacher networks learn better representations, b) the teacher is able to transfer *dark knowledge* (Hinton et al., 2014) in its predictions and c) the regularization effect of the soft labels in the KD loss (Yuan et al., 2020). Nonetheless, it is unclear how exactly student networks benefit from these soft labels. Recent work suggests that with KD, the student is unable to match the teacher on the training distribution but improves generalization on the test set (Stanton et al., 2021).

In trying to understand KD, a series of investigations have looked at regularization and have demonstrated that the success of both KD and label smoothing is due to a similar regularization effect of soft targets (Yuan et al., 2020; Zhang & Sabuncu, 2020). Label smoothing (LS) (Szegedy et al., 2016) is a popular technique for enhancing the generalization of a wide range of models without incurring additional computational costs. It encourages the model to treat each non-target class as equally likely for classification by using a uniform distribution to smooth the distribution of the hard labels. Although combining the uniform distribution with the original hard label is beneficial for regularisation, LS does not take into account the true relationships between different label categories like the way KD does. Zhang & Sabuncu (2020) demonstrate the importance of an instance

specific LS regularization. They demonstrate better performance compared to LS, but use a trained model to infer prior knowledge on the label space and thereby sacrifice some of the efficiency of LS.

In this work, we investigate *whether we can bridge the performance gap between LS and KD, while maintaining the efficiency of LS*. Motivated by this, we first revisit the conventional LS technique and generalize the uniform LS method to an instance-specific smoothing regularization framework. Within this framework, we demonstrate that both LS and KD can be interpreted as instances of a smoothing distribution. Next, we propose to learn the optimal smoothing function along with the model training, by devising a two-stage optimization problem. We solve the first-stage problem by giving a deterministic and interpretable solution and apply gradient based optimization to solve the second stage.

Our contributions can be summarized as follows:

- Our method, Pseudo-KD, provides a general framework, that improves upon conventional LS without an additional training cost.
- We justify our framework by demonstrating the new formulation unifies the conventional LS and KD objectives, which confirm their relationship from a different perspective.
- Based on our framework, we further propose to learn the smoothing distribution along with training by formulating it as a bi-level optimization problem and we derive a deterministic and interpretable solution.
- We conducted extensive experiments on image classification (CIFAR10 and CIFAR100) and natural language understanding (the GLUE benchmark) and show that our method outperforms label smoothing, and performs on par with KD while being significantly faster in training (approximately up to 3x for some settings).

## 2 RELATED WORK

We will provide a brief overview of KD and LS and *teacher-free* methods which reconcile these two.

**Knowledge Distillation.** KD came into prominence after the work of Hinton et al. (2015). Given access to a trained teacher model, assume that we want to train a student. Denote by $P_T(x)$ and $p_\theta(x)$ the teacher's and student's predictions respectively. For a classification problem, the total KD loss is defined as:

$$L = (1 - \alpha)H(q(x), p_\theta(x)) + \alpha D_{KL}(P_T(x), p_\theta(x)) \tag{1}$$

where $q(x)$ is the one-hot label, $H(q(x), p(x))$ is the cross-entropy loss between the student's output and the labels, $D_{KL}$ is the KL divergence and $\alpha$ is the scaling parameter. Note that we assume a temperature of 1 and omit it without loss of generality. KD training is equivalent to training with a smoothed label $\hat{P}(x)$ such that:

$$\hat{P}_T(x) = (1 - \alpha)q(x) + \alpha P_T(x) \tag{2}$$

**Label Smoothing** We observe that the loss in equation 2 is similar in form to the LS loss (Yuan et al., 2020). In LS the KL divergence is between the student output and a uniform distribution $U(k) = \frac{1}{K}$ where $K$ is the number of classes. Therefore we can define label smoothened label as:

$$\hat{P}_U(x) = (1 - \alpha)q(x) + \alpha U \tag{3}$$

Both LS and KD incorporate training a model with a smoothened label. KD tends to perform better as it leads to a more accurate student (Müller et al., 2019; Shen et al., 2021). However, LS is more efficient since it does not need a pre-trained teacher network. We note from equation 3 that a higher $\alpha$ can lead to a smoother label distribution and a lower $\alpha$ to a *peakier* label. Yuan et al. (2020) exploited this to propose a better LS loss to bridge the performance gap between KD and LS. Chen et al. (2021) note that training on high entropy label distributions can also lead to better robustness under adversarial attacks.

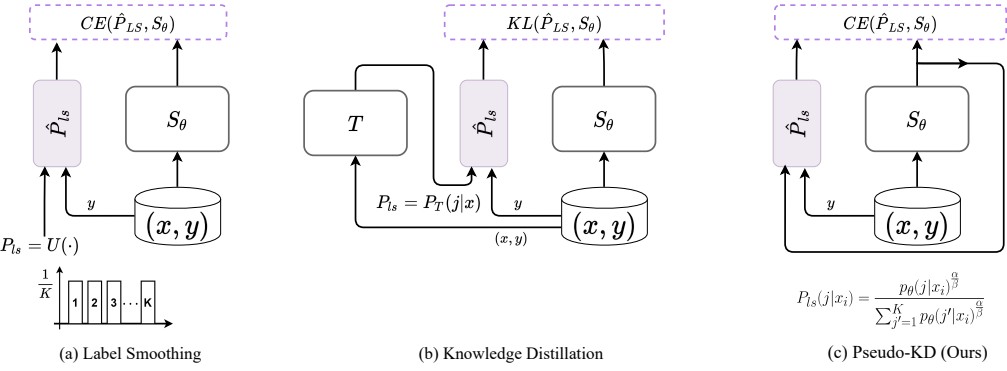

Figure 1: Our Pseudo-KD method is compared with the label smoothing and knowledge distillation techniques.

**Self Distillation** We can take advantage of distillation without necessarily using a cumbersome teacher. A student can improve generalization by learning for the output of a trained, identically parameterized model Furlanello et al. (2018); Yuan et al. (2020). Zhang et al. (2018) propose to train an ensemble of students without a static teacher, which benefits from learning and teaching among different students collaboratively. Another self-boosting direction is to apply regularization on the intermediate layers instead of predictive distributions. During the training stage, Zhang et al. (2019) introduce auxiliary classifiers on top of intermediate layers and penalize the difference between the predictions of these classifiers and the penultimate layer. Along this direction, Liu et al. (2020) further propose a top-down feature fusion process to handle the capacity gap between the intermediate layers with different depths. However, all these methods require extra parameterized modules for training which are abandoned at inference. In contrast, some recent works focus on boosting the performance with explicitly regularized soft targets. Zhang & Sabuncu (2020) interpret student-teacher training as an amortized maximum a posteriori estimation and derive an equivalence between self-distillation and instance-specific label smoothing. This analysis helped them to devise a regularization scheme, Beta Smoothing. However, they still use an extra model to infer a prior distribution on their smoothing technique during the training. Yun et al. (2020) introduce an additional regularization to penalize the predictive output between different samples of the same class. Ding et al. (2021) propose to capture the relation of classes by introducing decoupled labels table which increases $O(K \times K)$ space complexity. The concurrent work (Kim et al., 2021) utilizes the model trained at $i$-th epoch as the teacher to regularize the training at $(i+1)$-th epoch along with annealing techniques. The difference emerges as it still requires a separated model as teacher, or saving all soft labels generated at $i$-th epoch for the next epoch training which introduces $O(N \times K)$ space complexity. Our work has a different basis: we develop Pseudo-KD motivated by a principled starting point: to generalize the smoothing distribution to a general form with neither time nor space complexity increases during the training and inference.

## 3 METHODOLOGY

In this section, we derive our Pseudo-KD framework from the two-stage optimization perspective. First, we revisit the conventional uniform LS method and introduce a closed-form solution to our optimal instance-specific label smoothing. Then, we describe a teacher-free KD implementation under this formulation.

### 3.1 REVISITING LABEL SMOOTHING

Suppose that we have a $K$-class classification task to be learned by a neural network, $S_\theta(\cdot)$. Given a training set of $\{x_i, y_i\}_{i=1}^N$ samples where $x_i$ is a data sample and $y_i$ is the ground-truth label, the

model $S_\theta(\cdot)$ outputs the probability $p_\theta(j|x_i)$ of each class $j \in \{1, \ldots, K\}$:

$$p_\theta(j|x_i) = \frac{\exp(z_{i,j})}{\sum_{j'=1}^{K} \exp(z_{i,j'})}, \tag{4}$$

where $z_{i,j} = [S_\theta(x_i)]_j$ is the logit for $j$-th class of input $x_i$.

A general label smoothing can be formally written as:

$$\hat{P}_{ls}(j|x_i) = \begin{cases} 1 - \alpha + \alpha \cdot P_{ls}(j|x_i), & j = k, \text{ where k is the ground-truth class;} \\ \alpha \cdot P_{ls}(j|x_i), & j \neq k \end{cases}, \tag{5}$$

where $\hat{P}_{ls}$ is a smoothed label, $P_{ls}$ is a smoothing distribution, and $\alpha$ is a hyperparameter that determines the amount of smoothing. If $\alpha = 0$, we obtain the original one-hot encoded label. For the original label smoothing method, the probability $P_{ls}(\cdot|x)$ is independent on the sample $x$ and is taken to be the uniform distribution $P_{ls}(j) = U(j)$, However, the training can benefit from instance-specific regularization. (Yuan et al., 2020; Zhang & Sabuncu, 2020). In this work, we consider the general form of LS $\hat{P}_{ls}(\cdot|x_i)$ which is instance-specific and not necessarily uniform.

Let us consider the Cross Entropy (CE) loss with the smoothed labels:

$$\mathcal{L}_\theta(P_{ls}) = \frac{1}{N} \sum_{i=1}^{N} \left[ -\sum_{j=1}^{K} \hat{P}_{ls}(j|x_i) \log p_\theta(j|x_i) \right] = \frac{1}{N} \sum_{i=1}^{N} \mathbb{E}_{\hat{P}_{ls}} \left[ -\log p_\theta(j|x_i) \right]. \tag{6}$$

Note that computing the weighted sum of negative log-likelihood of each probability of label can be viewed as taking expectation of the negative log-likelihood over label space under a certain distribution $\hat{P}_{ls}$. We modify this loss by adding a Kullback–Leibler (KL) divergence term $D_{\mathrm{KL}}(P_{ls}(\cdot|x_i)\|U(\cdot))$ into Eq. 6 which encourages the sample-wise smoothing distribution $P_{ls}(\cdot|x_i)$ to be close to the uniform distribution.

$$\mathcal{R}_\theta(P_{ls}) = \frac{1}{N} \sum_{i=1}^{N} \left[ \mathbb{E}_{\hat{P}_{ls}} \left[ -\log p_\theta(j|x_i) \right] + \beta D_{\mathrm{KL}}(P_{ls}(\cdot|x_i)\|U(\cdot)) \right], \tag{7}$$

where $\beta$ is a hyper-parameter. The instance-specific smoothing $P_{ls}(\cdot|x_i)$ can be viewed as the prior knowledge over the label space for a sample $x_i$. This KL divergence term can be understood as a measure of the 'smoothness' or 'overconfidence' of each training label. We choose a Uniform distribution to constrain $P_{ls}$ in the KL term because we would like $P_{ls}$ contain more information on non-target classes. It plays a role similar to temperature in KD and controls the sharpness of the distribution $P_{ls}$. On the one hand side, in case of KD, it regulates the amount of prior knowledge we inject in the soft label. On the other hand side, it reduce the overconfidence of the trained network by increasing the entropy of soft labels (a phenomenon studied by Zhang & Sabuncu (2020)).

Next we show the post-hoc interpretability of our new formulation. The following two Remarks discuss the relationship of our objective with LS and KD.

**Remark.** *When $P_{ls}(\cdot|x_i)$ is taken to be the uniform distribution $U(\cdot)$ for any $x_i$, the objective in Eq. 7 reduces back to the one in Eq. 6 since $D_{\mathrm{KL}}\big(U(\cdot)\|U(\cdot)\big) = 0$.*

**Remark.** *When $P_{ls}(\cdot|x_i)$ is taken to be $P_T(\cdot|x_i)$ which is the output probability of a teacher model $T(\cdot)$ for any $x_i$, our objective can be converted to the conventional KD objective with temperature $\tau = 1$ and plus a constant $\alpha \log K$*

$$\mathcal{R}_\theta(P_T) = \frac{1}{N} \sum_{i=1}^{N} \left[ \mathbb{E}_{\hat{P}_T} \left[ -\log p_\theta(j|x_i) \right] + \alpha D_{\mathrm{KL}}(P_T(\cdot|x_i)\|U(\cdot)) \right], \tag{8}$$

$$= \frac{1}{N} \sum_{i=1}^{N} \left[ (1-\alpha)\mathcal{L}_{CE}(y_i, p_\theta(\cdot|x_i)) + \alpha D_{\mathrm{KL}}(P_T(\cdot|x_i)\|p_\theta(\cdot|x_i)) \right] + \alpha \log K,$$

*where $\hat{P}_T(j|x_i) = (1 - \alpha + \alpha \cdot P_T(k|x_i))$ for $j = k$ the ground truth, and $\hat{P}_T(j|x_i) = \alpha \cdot P_T(j|x_i)$ for $j \neq k$. Our objective function Eq. 7 demonstrates that the both objective of KD and LS can*

*be rewritten as an expectation of negative log-likelihood w.r.t. a transformed distribution $\hat{P}_T$ or $U(\cdot)$ plus a KL term between the smoothing distribution and uniform distribution.* *This is not only to confirm previous findings from a different perspective but also justify why our framework is reasonable.*

*Our objective gives a unified formulation for instance-specific label smoothing in a general form. It is easy to bridge the objective of label smoothing and knowledge distillation by converting our objective to KD or LS via replacing the $P_{ls}(\cdot|x_i)$ in Eq. 7 with $P_T(\cdot|x_i)$ or $U(\cdot)$, respectively. There is an inherent relation among these methods. KD and LS, in particular, can be interpreted as instances of our method with a fixed smoothing distribution.*

### 3.2 Pseudo-KD via two-stage Optimization

The choice of the distribution $P_{ls}$ determines the label regularization method. As described above, in KD the smoothing distribution is an output of the pre-trained teacher model, and in LS it is a uniform distribution. Generally speaking, the optimal smoothing distribution is unknown, and ideally, we would like to learn the optimal $P_{ls}$. In this regard, we set up the following two-stage optimization problem:

$$\min_{\theta} \mathcal{R}_{\theta}(P_{ls}^*),$$
$$\text{subject to } P_{ls}^* = \arg\min_{P_{ls}} \mathcal{R}_{\theta}(P_{ls}). \tag{9}$$

This optimization setting, also called bilevel optimization (Colson et al., 2007), is strongly NP hard (Jeroslow, 1985) so getting an exact solution is difficult. Luckily, in our case the inner optimization loop has a closed-form solution. The following Theorem 1 gives the explicit formulation of $\mathcal{R}_{\theta}(P_{ls}^*)$. In particular, we will show that $\mathcal{R}_{\theta}(P_{ls})$ is a convex function w.r.t $P_{ls}$, and we will obtain the optimal solution $P_{ls}^* = \arg\min_{P_{ls}} \mathcal{R}_{\theta}(P_{ls})$ by using a Lagrangian multiplier. For the details of this derivation please refer to Appendix A.

**Theorem 1.** *The solution to the inner loop of optimization in Eq. 9 is given by:*

$$P_{ls}^*(j|x_i) = \frac{p_{\theta}(j|x_i)^{\frac{\alpha}{\beta}}}{\sum_{j'=1}^{K} p_{\theta}(j'|x_i)^{\frac{\alpha}{\beta}}}, \tag{10}$$

*where $p_{\theta}(j|x_i)$ is the output probability of $j$-th class of model $S_{\theta}(\cdot)$, $\alpha$ is the smoothing coefficient defined in Eq. 5, and $\beta$ is defined in Eq. 7.*

As a result, we reduced the two-stage optimization problem in Eq. 9 to a regular single-stage minimization:

$$\min_{\theta} \mathcal{R}_{\theta}(P_{ls}^*), \tag{11}$$

where

$$\mathcal{R}_{\theta}(P_{ls}^*) = \frac{1}{N} \sum_{i=1}^{N} \left[ \sum_{j=1}^{K} \hat{P}_{ls}^*(j|x_i) \cdot (-\log p_{\theta}(j|x_i)) + \beta P_{ls}^*(j|x_i) \log(K \cdot P_{ls}^*(j|x_i)) \right], \tag{12}$$

$P_{ls}^*(j|x_i)$ is given in Theorem 1, and $\hat{P}_{ls}^*(j|x_i)$ is defined in Eq. 5. Note that the solution $P_{ls}^*$ is deterministic and interpretable. Moreover, we show the post-hoc interpretability of the solution in the two remarks below by demonstrating the relation of the solution with LS and self-distillation methods.

**Remark.** *When $\beta$ is extremely large, $P_{ls}^*$ will be close to the Uniform distribution, and our objective function will be equivalent to optimizing the CE loss with a Uniform LS regularizer. When $\beta$ is extremely small, the smoothing distribution collapses to the one-hot vector.*

**Remark.** *There is an intrinsic connection between the $P_{ls}^*$ distribution and generating softmax outputs with a temperature factor. Specifically, when $\beta = \alpha \cdot \tau$, we could have*

$$P_{ls}^*(j|x_i) = \frac{\exp(\frac{z_{i,j}}{\tau})}{\sum_{j'=1}^{K} \exp(\frac{z_{i,j'}}{\tau})} \tag{13}$$

*Since the smoothing distribution in this case becomes the temperature smoothed output of the model, our Pseudo-KD technique can be considered as a generalized version of the self-distillation method [1].*

To summarize, our method can be expressed as an alternating two-stage process. We generate optimal smoothed labels $P_{ls}^*$ using Theorem 1 in the first stage. Then, in the second phase, we fix the $P_{ls}^*$ to compute loss $\mathcal{R}_\theta(P_{ls}^*)$ and update the model parameters. We execute our method in an online manner, which eliminates the need for additional memory or storage for the parameters or outputs of the pre-trained model. The online training process is shown in Algorithm 1.

---

**Algorithm 1** Online Training with Pseudo-KD

---

**Input:** Training set $\{(x_i, y_i)\}|_{i=1}^N$, batch size $n$, number of training epochs $T$, learning rate $\eta$;

1: **for** $i \leftarrow 1$ **to** $T$ **do**
2:     **for** $t \leftarrow 1$ **to** $N/n$ **do**
3:         Randomly sample a mini-batch $S = \{(x_i, y_i)\}|_{i=1}^n$ from training set;
4:         Compute the $\hat{P}_{ls}$ according to $\hat{P}_{ls}^*$ solution for the mini-batch data;      ▷ 1st Stage
5:         Update the model $\theta_{t+1} = \theta_t - \eta \nabla_\theta R_\theta(\hat{P}_{ls}^*, S)$;      ▷ 2nd Stage
6:     **end for**
7: **end for**

---

## 4 Experiments

In this section, we evaluate the performance of our proposed Pseudo-KD method. Experiments are conducted on both the image classification tasks CIFAR-10 and CIFAR-100 (Krizhevsky et al., 2009) for the ResNet-based Model with of various sizes (parameters) and the General Language Understanding Evaluation (GLUE) tasks (Wang et al., 2019b) with the RoBERTa-based model (Liu et al., 2019).

### 4.1 Experiments on Image Classification

**Data.** CIFAR is a benchmark dataset for image classification (Krizhevsky et al., 2009). We use both CIFAR-10 and CIFAR-100, which have images 50,000 training samples and 10,000 test samples divided into 10 and 100 different object classes, respectively. The validation set is made up of 10% of the training data.

**Experimental settings.** We evaluated our method on different model architectures including MobileNetV2 (Sandler et al., 2018), ShuffleNetV2 (Ma et al., 2018), ResNeXt29(8×64d) (Xie et al., 2017), ResNet18, ResNet50 and ResNet101 (He et al., 2016). We follow standard data augmentation schemes: random crop and horizontal flip to augment the original training images. The models are trained for 200 epochs, with a batch size of 128 for MobileNetV2, ShuffleNetV2 and ResNet18, and 64 for ResNeXt29 (8×64d), ResNet50 and ResNet101. For optimization we used stochastic gradient descent with a momentum of 0.9, and weight decay set to 5e-4. The learning rate starts at 0.1 and is then divided by 5 at epochs 60, 120 and 160. All experiments are repeated 5 times with random initialization. For the KD experiments (KD-Shuffle and KD-29), we used ShuffleNetv2 and ResNeXt29 as the teachers. All teacher models are trained from scratch and picked based on their best accuracy. To explore the best hyper-parameters, we conduct a grid search over parameter pools. We explore $\{0.1, 0.2, 0.5, 0.9\}$ for $\alpha$ and $\{5, 10, 20, 40\}$ for KD temperature. For Pesudo-KD, we also perform explore $\{1.5, 2.5, 4\}$ for $\beta$. The best hyper-parameters can be found in Appendix C.

We first compare our proposed Pseudo-KD method with uniform label smoothing and conventional KD with different teachers. All experiments are repeated 5 times with different random initialization.

**Results** Tab. 4.1 shows the test accuracy for each method using various network architectures. Our method significantly improves classification performance than conventional LS on all six models with over 0.85 or 0.35 points higher accuracy on average for CIFAR100 and CIFAR10, respectively.

---

[1]The derivation can be found in Appendix B.

Table 1: Accuracy on CIFAR 100 and CIFAR 10 test set. "mean ($\pm$ std)" are reported on 5 runs.

|  |  | Base | LS | KD-Shuffle | KD-29 | Pseudo-KD (Ours) |
|---|---|---|---|---|---|---|
| CIFAR100 | ShuffleNetv2 | 70.45 ($\pm$0.22) | 70.78 ($\pm$0.35) | **72.09** ($\pm$0.21) | 71.97 ($\pm$0.26) | 71.95 ($\pm$0.26) |
|  | MobileNetv2 | 67.98 ($\pm$0.13) | 68.69 ($\pm$0.33) | 70.93 ($\pm$0.30) | 70.99 ($\pm$0.16) | **71.05** ($\pm$0.35) |
|  | ResNet18 | 76.92 ($\pm$0.11) | 77.67 ($\pm$0.27) | 77.73 ($\pm$0.10) | 77.78 ($\pm$0.22) | **78.10** ($\pm$0.15) |
|  | ResNeXt29 | 81.07 ($\pm$0.30) | 82.02 ($\pm$0.17) | 81.86 ($\pm$0.16) | **82.24** ($\pm$0.18) | 82.19 ($\pm$0.16) |
|  | ResNet50 | 79.28 ($\pm$0.12) | 79.16 ($\pm$0.42) | 79.51 ($\pm$0.18) | 79.65 ($\pm$0.36) | **79.76** ($\pm$0.31) |
|  | ResNet101 | 78.81 ($\pm$0.35) | 79.11 ($\pm$0.33) | 79.30 ($\pm$0.20) | **79.58** ($\pm$0.21) | 79.45 ($\pm$0.25) |
| CIFAR10 | ShuffleNetv2 | 91.32 ($\pm$0.27) | 91.67 ($\pm$0.24) | **92.30** ($\pm$0.09) | 92.26 ($\pm$0.15) | 91.99 ($\pm$0.22) |
|  | MobileNetv2 | 90.55 ($\pm$0.20) | 90.82 ($\pm$0.12) | 91.46 ($\pm$0.25) | 91.52 ($\pm$ 0.15) | **91.53** ($\pm$0.13) |
|  | ResNet18 | 94.82 ($\pm$0.09) | 95.02 ($\pm$0.15) | 95.05 ($\pm$0.05) | **95.28** ($\pm$0.08) | 95.21 ($\pm$0.09) |
|  | ResNeXt29 | 95.72 ($\pm$0.11) | 95.71 ($\pm$0.09) | 95.75 ($\pm$0.13) | **96.06** ($\pm$0.07) | 95.82 ($\pm$0.14) |
|  | ResNet50 | 95.15 ($\pm$0.11) | 95.10 ($\pm$0.09) | 95.21 ($\pm$0.08) | **95.52** ($\pm$0.08) | 95.43 ($\pm$0.16) |
|  | ResNet101 | 95.25 ($\pm$0.13) | 94.98 ($\pm$0.19) | 95.29 ($\pm$0.07) | **95.62** ($\pm$0.07) | 95.44 ($\pm$0.15) |

This indicates its adaptability to different networks. Compared to KD methods, our method can give comparable or better performance, while being much more efficient in training. This is because our method could be implemented in an online manner, which only need one forward pass as Base or LS methods. Furthermore, our method does not necessitate the use of a pre-trained teacher, which saves computational overhead.

We also observe that lightweight models like ShuffleNetv2 and MobileNetv2 benefit more from LS, KD and our method than complex models (ResNeXt29 and ResNet101). Furthermore, as a teacher, using a model with a low capacity can provide comparable performance for improvements for small models to using a large model. Specifically, by distilling three low-capacity models (ShuffleNetv2, MobileNetv2, and ResNet18) and using ShuffleNet as the teacher, all three models can achieve comparable performance, with ShuffleNet outperforming the others.

Table 2: Comparison between different Teacher-free methods. Average test accuracy and training time are reported. The training time is measured on a single NVIDIA V100 GPU.

|  | CIFAR100 (Acc. & Time) | | | | | | CIFAR 10 (Acc. & Time) | | | | | |
|---|---|---|---|---|---|---|---|---|---|---|---|---|
|  | MobileNetv2 | | ResNet18 | | ResNet50 | | MobileNetv2 | | ResNet18 | | ResNet50 | |
| Base | $67.98_{\pm0.13}$ | 1.1h | $76.92_{\pm0.11}$ | 1.1h | $79.28_{\pm0.12}$ | 3.6h | $90.55_{\pm0.20}$ | 1.1h | $94.82_{\pm0.09}$ | 1.1h | $95.15_{\pm0.11}$ | 3.6h |
| KD-29 | $70.99_{\pm0.16}$ | 4.1h | $77.78_{\pm0.22}$ | 4.2h | $79.65_{\pm0.36}$ | 14.4h | $91.52_{\pm0.15}$ | 4.1h | $95.28_{\pm0.08}$ | 4.2h | $95.52_{\pm0.08}$ | 14.4h |
| LS | $68.69_{\pm0.33}$ | 1.1h | $77.67_{\pm0.27}$ | 1.1h | $79.16_{\pm0.42}$ | 3.6h | $90.82_{\pm0.12}$ | 1.1h | $95.02_{\pm0.15}$ | 1.1h | $95.10_{\pm0.09}$ | 3.6h |
| CS-KD | $70.36_{\pm0.24}$ | 1.2h | $77.95_{\pm0.10}$ | 1.4h | $79.33_{\pm0.20}$ | 3.9h | $91.17_{\pm0.11}$ | 1.2h | $94.90_{\pm0.08}$ | 1.4h | $95.25_{\pm0.09}$ | 3.9h |
| TF-reg | $70.08_{\pm0.34}$ | 1.1h | $77.91_{\pm0.12}$ | 1.3h | $79.19_{\pm0.34}$ | 3.7h | $90.97_{\pm0.12}$ | 1.1h | $95.05_{\pm0.10}$ | 1.3h | $95.12_{\pm0.10}$ | 3.7h |
| Beta-LS | $70.45_{\pm0.25}$ | 1.6h | $77.83_{\pm0.10}$ | 1.5h | $79.55_{\pm0.25}$ | 4.8h | $90.89_{\pm0.09}$ | 1.6h | $94.87_{\pm0.11}$ | 1.8h | $95.22_{\pm0.08}$ | 4.5h |
| KR-LS | $70.12_{\pm0.23}$ | 1.1h | $77.82_{\pm0.27}$ | 1.1h | $79.48_{\pm0.18}$ | 3.6h | $90.67_{\pm0.22}$ | 1.1h | $94.76_{\pm0.15}$ | 1.1h | $95.30_{\pm0.15}$ | 3.6h |
| PS-KD | $70.94_{\pm0.23}$ | 1.2h | **$78.36_{\pm0.17}$** | 1.2h | **$79.83_{\pm0.20}$** | 3.6h | **$91.56_{\pm0.10}$** | 1.2h | $95.14_{\pm0.13}$ | 1.2h | $95.39_{\pm0.18}$ | 3.6h |
| Pseudo-KD | **$71.05_{\pm0.35}$** | 1.1h | $78.10_{\pm0.15}$ | 1.1h | $79.76_{\pm0.31}$ | 3.6h | $91.53_{\pm0.13}$ | 1.1h | **$95.21_{\pm0.09}$** | 1.1h | **$95.43_{\pm0.16}$** | 3.6h |

Next, we conduct a series of experiments on three models to compare our approach with other methods without requiring any pre-trained models (except for KD-29 reported as a reference). All experiments are repeated 5 times with different random initialization. The baseline methods include, CS-KD which distill the soft output between different samples of the same class to student (Yun et al., 2020). TF-reg which regularize the training with a manually designed teacher (Yuan et al., 2020) and Beta-LS which leveraging a model with same capacity to learn instance-specific prior over the smoothing distribution (Zhang & Sabuncu, 2020). We follow their best hyper-parameter settings. The time for all baselines is measured based on its original implementations. [2345]

Tab. 2 shows the test accuracy and training time for 200 epoch of different methods on three models. It can be seen that our method consistently improves the classification performance on both

[2]https://github.com/alinlab/cs-kd

[3]https://github.com/yuanli2333/Teacher-free-Knowledge-Distillation

[4]https://github.com/ZhiluZhang123/neurips_2020_distillation

[5]https://github.com/lgcnsai/PS-KD-Pytorch

lightweight and complex models, which indicates its robustness to different networks. Besides, it shows the training time of our method is equivalent to Base or LS methods, which show its stable efficiency over other baselines such as Beta-LS, which still requires a separated model to output a learned prior over the smoothing distribution. We further compare two recent works KR-LS (Ding et al., 2021) and PS-KD (Kim et al., 2021). Our method achieves comparable or better performance than PSKD and consistently better than KR-LS. We have to mention the implementation of KR-LS and PS-KD increase space complexity $O(K \times K)$ and $O(N \times K)$ for class-wise and sample-wise smoothing labels, respectively. However, our method neither increases time nor space complexity during the training.

## 4.2 RESULTS ON NATURAL LANGUAGE UNDERSTANDING

**Data.** We run experiments on 7 classification tasks from the GLUE benchmark (Wang et al., 2019b). These datasets can be broadly divided into 3 families of problems. Single set tasks which include linguistic acceptability (CoLA) and sentiment analysis (SST-2), similarity and paraphrasing tasks (MRPC and QQP), and inference tasks which include Natural Language Inference (MNLI and RTE) and Question Answering (QNLI). Following prior works (Jafari et al., 2021; Rashid et al., 2021), we report Spearman correlation for STS-B, Matthews correlation for CoLA, and accuracy for the other tasks for the development set. We report "F1" for the QQP and MRPC and the corresponding test sets.

**Experimental Setup.** We choose the pre-trained DistilRoBERTa (6-layer) (Sanh et al., 2019) as the base model for all our experiments. For KD experiments, we use a RoBERTa-Large model (24-layer) (Liu et al., 2019) fine-tuned on each of the tasks as the corresponding teacher model. All models are trained with the AdamW optimizer with the default settings Loshchilov & Hutter (2019). To explore the best hyper-parameters, we conduct a grid search over the learning rate $\in \{1e-5, 2e-5, 3e-5\}$, batch size $\in \{8, 16, 32, 64\}$, $\alpha \in \{0.1, 0.2, 0.5\}$ and KD temperature $\in \{1, 5, 10\}$. Moreover, for Pesudo-KD, we perform a search over three $\beta$ values in $\{1.5, 2, 4\}$. The best hyper-parameters can be found in Appendix. We repeat our experiments 3 times with different random initialization.

Table 3: GLUE results on Dev set.

|  | CoLA | MRPC | RTE | SST-2 | QNLI | QQP | MNLI | Avg. |
|---|---|---|---|---|---|---|---|---|
| Teacher | 68.1 | 91.9 | 86.3 | 96.4 | 94.6 | 91.5 | 89.9 | 88.4 |
| Student | $61.0_{\pm0.28}$ | $86.5_{\pm0.34}$ | $69.1_{\pm2.25}$ | $92.7_{\pm0.06}$ | $91.2_{\pm0.22}$ | $91.3_{\pm0.21}$ | $83.4_{\pm0.19}$ | 82.2 |
| w/ KD | $61.4_{\pm0.54}$ | $86.1_{\pm0.31}$ | $70.2_{\pm0.17}$ | $92.6_{\pm0.11}$ | $\mathbf{91.5}_{\pm0.14}$ | $\mathbf{91.8}_{\pm0.02}$ | $\mathbf{84.4}_{\pm0.24}$ | 82.6 |
| w/ LS | $60.3_{\pm0.54}$ | $86.4_{\pm0.42}$ | $70.3_{\pm1.79}$ | $92.8_{\pm0.05}$ | $91.3_{\pm0.18}$ | $91.5_{\pm0.10}$ | $83.8_{\pm0.10}$ | 82.3 |
| w/ Pseudo-KD (Ours) | $\mathbf{61.4}_{\pm0.37}$ | $\mathbf{86.9}_{\pm0.29}$ | $\mathbf{70.4}_{\pm1.06}$ | $\mathbf{93.0}_{\pm0.15}$ | $\mathbf{91.5}_{\pm0.14}$ | $91.5_{\pm0.23}$ | $84.1_{\pm0.21}$ | $\mathbf{82.7}$ |

Table 4: GLUE results on Test set for best model on dev sets.

|  | CoLA | MRPC | RTE | SST-2 | QNLI | QQP | MNLI | Avg. |
|---|---|---|---|---|---|---|---|---|
| Student | 50.5 | 88.9 | 63.5 | 92.8 | 90.8 | 88.9 | 83.7 | 79.8 |
| w/ KD | 51.5 | 88.6 | 63.3 | 93.0 | 91.5 | 89.2 | 84.5 | 80.2 |
| w/ LS | 49.6 | 88.6 | 64.6 | 93.0 | 90.8 | 89.0 | 83.8 | 79.9 |
| w/ Pseudo-KD (Ours) | 51.2 | 88.9 | 63.2 | 93.5 | 91.0 | 89.2 | 84.0 | 80.1 |

**Results** Table 3 presents the dev set results on GLUE. We present the average result for each experiment, over three runs, as well as the standard deviation. As is standard on the GLUE benchmark, we present an average result across the datasets as well. We compare Pseudo-KD to simple fine-tuning, LS and KD. We observe that our method is better on all datasets compared to fine-tuning and at par (QQP only) or better on all datasets compared to LS. We are comparable on average to

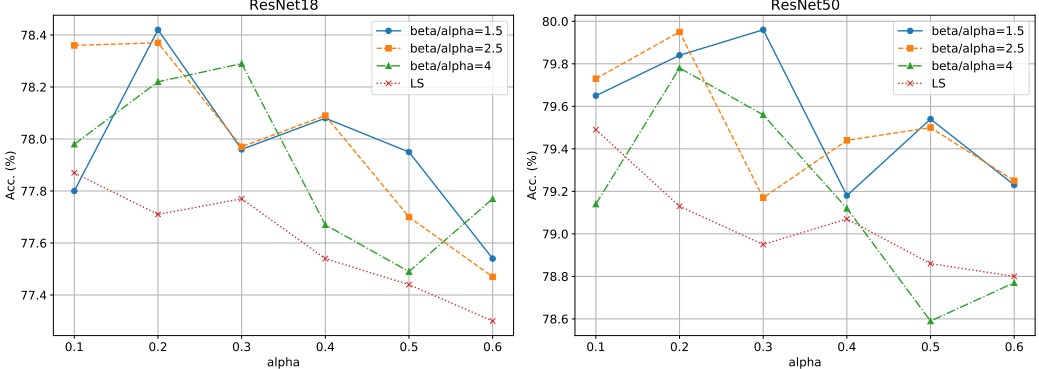

Figure 2: Effect of different combinations of $\alpha$ and $\beta$ on the performance of ResNet18 and ResNet50. For illustration, all values of $\beta$ are divided by corresponding $\alpha$.

KD, with KD doing better on MNLI and QQP and Pseudo-KD doing better o CoLA, MRPC, RTE and SST-2.

Furthermore, Table 4 shows the GLUE leaderboard test results. We chose the model with the best dev results for each method and submitted it to the official GLUE benchmark to get the test results. We see a similar trend and observe that Pseudo-KD is better than LS and fine-tuning on all datasets except RTE. On average the results are comparable to KD as before. KD is better on CoLA, RTE, QNLI and MNLI and Pseudo KD is better on MRPC, SST-2 and at par on QQP. We observe that on average all the models are close in performance, with an average difference of 0.4 between the best and the worst model.

### 4.3    EFFECT OF HYPER-PARAMETER $\alpha$ AND $\beta$

We conducted ablation tests to probe how the $\alpha$ and $\beta$ hyper-parameters affects the performance of our method. We also compared our results to LS while choosing the same $\alpha$ for both. Specifically, we chose $\alpha \in \{0.1, 0.2, \dots, 0.6\}$. For Pseudo-KD, we also verified the influence of $\beta/\alpha \in \{1.5, 2.5, 4\}$. Following the discussion on equation 13 we observe that $\beta/\alpha$ can be interpreted as the temperature in self-distillation.

We plot the accuracy on CIFAR100 for two architectures (ResNet18 and ResNet50) as shown in Fig. 2. The plot shows that our method can consistently outperform the LS method across different $\alpha$ values even when varying the $\beta/\alpha$ parameter. The results are consistent for both architectures. As is expected, the accuracy of both LS and our method gradually decreases when $\alpha$ increases from 0.1 to 0.6. It is interesting that even at an $\alpha$ of 0.6 the drop in accuracy is less than $1\%$ of the peak value.

We observe that a *temperature* value between 1.5 and 2.5 is the best to obtain optimal performance.

## 5    CONCLUSION

Our aim in this work is to fill the accuracy gap between KD and LS techniques, while maintaining the training efficiency of LS. We proposed learning an instance-specific label smoothing regularization simultaneously with training our model on the target. We began by revisiting the traditional LS method and introduced our objective function by substituting the uniform distribution with a general, instance-specific, discrete distribution. Within the new objective, we explained the relationship between the LS and KD. Particularly, our method can be interpreted as an online version of self-distillation. Then, using a two-stage optimization approach, we obtained an approximation for the optimal smoothing function. We conducted extensive experiments to compare our model with LS, KD and various teacher-free methods on popular CV and NLU benchmarks and showed the effectiveness and efficiency of our technique.

We want to extend this research in two directions in the future. First we want to improve our algorithm further and explore replacing the alternating approach to two-stage optimization with a more complex algorithm to reach closer to the optimal smoothing.

Next we want to explore practical applications where our method can replace KD. One interesting application is the pre-training of compressed language models. Typically, the best compressed models are pre-trained using distillation from a much larger teacher which makes the training process cumbersome and computationally intensive. It would be interesting to explore if Pseudo-KD can achieve comparable performance to KD in that setting.

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

## A  DERIVATION OF THE PROOF FOR THEOREM 1

*Proof.* First, observe that the function $\mathcal{R}_\theta$ defined in Eq. 7 is a convex combination of $N$ nonnegative functions $\mathcal{R}_i = \mathbb{E}_{\hat{P}_{ls}}\left[-\log p_\theta(j|x_i)\right] + \beta D_{\mathrm{KL}}(P_{ls}(\cdot|x_i)\|U(\cdot))$, for $i = 1, \ldots, N$. We will show that each of $\mathcal{R}_i$ is a convex function of the components of simplex $P_{ls}(\cdot|x_i)$ by computing the Hessian matrix of $\mathcal{R}_i$ with respect to $P_{ls}(\cdot|x_i)$:

$$
\mathbf{H}(R_i(P_{ls}(1), \cdots, P_{ls}(K))) = \begin{bmatrix} \frac{\partial^2 \mathcal{R}_i}{\partial P_{ls}(1)^2}, & \frac{\partial^2 \mathcal{R}_i}{\partial P_{ls}(1)\partial P_{ls}(2)}, & \cdots, & \frac{\partial^2 \mathcal{R}_i}{\partial P_{ls}(1)\partial P_{ls}(K)} \\ \frac{\partial^2 \mathcal{R}_i}{\partial P_{ls}(2)\partial P_{ls}(1)}, & \frac{\partial^2 \mathcal{R}_i}{\partial P_{ls}(2)^2}, & \cdots, & \frac{\partial^2 \mathcal{R}_i}{\partial P_{ls}(2)\partial P_{ls}(K)} \\ \vdots, & \vdots, & \ddots, & \vdots \\ \frac{\partial^2 \mathcal{R}_i}{\partial P_{ls}(3)\partial P_{ls}(1)}, & \frac{\partial^2 \mathcal{R}_i}{\partial P_{ls}(3)\partial P_{ls}(2)}, & \cdots, & \frac{\partial^2 \mathcal{R}_i}{\partial P_{ls}(K)^2} \end{bmatrix} \tag{14}
$$

$$
= \mathrm{diag}\left(\frac{\beta}{P_{ls}(1)}, \frac{\beta}{P_{ls}(2)}, \cdots, \frac{\beta}{P_{ls}(K)}\right)
$$

When $\beta$ is greater than zero, the Hessian is positive definite. Therefore, each $R_i$ is a convex function of the componets of $P_{ls}(\cdot|x_i)$. As a result, $\mathcal{R}_\theta$ is a convex function of the collection of components of every simplex $P_{ls}(\cdot|x_i)$.

For simplicity, we derive the global optimum solution for each $\mathcal{R}_i$ with a Lagrangian multiplier:

$$
L_i(P_{ls}, \lambda_L) = \sum_{j'=1}^{K} \left[ -\hat{P}_{ls}(j') \log p(j') + \beta \cdot P_{ls}(j') \log \frac{P_{ls}(j')}{1/K} \right] + \lambda_L \left( \sum_{j'=1}^{K} P_{ls}(j') - 1 \right),
$$

where we omit the dependency on $x_i$ to simplify the notation. We set the corresponding gradients equal to 0 to obtain the global optimum for $j = 1, \ldots, K$.

$$
\frac{\partial L_i}{\partial P_{ls}(j)} = -\alpha \log p(j) + \beta \cdot \log P_{ls}(j) + \beta + \beta \log K + \lambda_L = 0 \tag{15}
$$

$$
P_{ls}^*(j) = \exp\left(\frac{\alpha}{\beta} \log p(j)\right) \cdot \exp\left(\frac{-\beta - \beta \log K - \lambda_L}{\beta}\right)
$$

$$
= \exp\left(\frac{\alpha}{\beta} \log p(j)\right) \cdot C_{ls}
$$

Since $\sum_{j'=1}^{K} P_{ls}(j') = 1$, we have

$$\sum_{j'=1}^{K} P_{ls}(j') = \sum_{j'=1}^{K} \exp(\frac{\alpha}{\beta} \log p(j')) \cdot \exp(\frac{-\beta - \beta \log K - \lambda_L}{\beta}) = 1 \tag{16}$$

$$C_{ls} = \exp(\frac{-\beta - \beta \log K - \lambda_L}{\beta}) = \frac{1}{\sum_{j'=1}^{K} \exp(\frac{\alpha}{\beta} \log p(j'))}$$

So the optimal $P_{ls}^*(j)$ is given by the formula:

$$P_{ls}^*(j) = \frac{\exp(\frac{\alpha}{\beta} \log p(j))}{\sum_{j'=1}^{K} \exp(\frac{\alpha}{\beta} \log p(j'))} = \frac{p(j)^{\alpha/\beta}}{\sum_{j'=1}^{K} p(j')^{\alpha/\beta}}. \tag{17}$$

$\square$

## B    DERIVATION FROM OPTIMAL SMOOTHING TO SOFTMAX OUTPUT WITH TEMPERATURE

When $\beta = \alpha \cdot \tau$, we could have

$$P_{ls}^*(c|x_i) = \frac{p_\theta(c|x_i)^{\frac{1}{\tau}}}{\sum_j p_\theta(j|x_i)^{\frac{1}{\tau}}} = \frac{(\frac{e^{z_{i,c}}}{\sum_m e^{z_{i,m}}})^{\frac{1}{\tau}}}{\sum_j (\frac{e^{z_{i,j}}}{\sum_m e^{z_{i,m}}})^{\frac{1}{\tau}}}$$

$$= \frac{(e^{z_{i,c}})^{\frac{1}{\tau}}}{\sum_j (e^{z_{i,j}})^{\frac{1}{\tau}}} = \frac{e^{\frac{z_{i,c}}{\tau}}}{\sum_j e^{\frac{z_{i,j}}{\tau}}} \tag{18}$$

This is the commonly used soft label in KD approach.

## C    HYPER-PARAMETERS FOR PSEUDO-KD

| Model | $\alpha$ | $\beta$ | batch size | lr |
|---|---|---|---|---|
| CIFAR100 | 0.2 | 1.5 | - | - |
| CIFAR10 | 0.2 | 2.5 | - | - |
| CoLA | 0.1 | 2 | 32 | 1e-5 |
| RTE | 0.1 | 2 | 16 | 2e-5 |
| SST-2 | 0.1 | 5 | 64 | 2e-5 |
| MRPC | 0.1 | 2 | 8 | 2e-5 |
| QNLI | 0.1 | 4 | 64 | 2e-5 |
| QQP | 0.1 | 4 | 32 | 2e-5 |
| MNLI | 0.1 | 2 | 32 | 2e-5 |

