# OpenReview forum: "Pseudo Knowledge Distillation: Towards Learning Optimal Instance-specific Label Smoothing Regularization"
_ICLR.cc/2022/Conference — ICLR 2022 Submitted_

### Official Review · Reviewer_JBRd · 2021-11-01

**Correctness:** 3
**Technical Novelty And Significance:** 2
**Empirical Novelty And Significance:** 2
**Recommendation:** 5
**Confidence:** 3

**Main Review:**

The authors propose a self-distillation approach that is derived from a series of observation made regarding how LS and KD work, linking in the two together in the process.

Pro:
+ The approach and the mathematical proofs provided are sound
+ The two step training process is fast and matches or outperform the vanilla KD baseline.
+ The performance is tested on both vision and language data showing that the approach generalizes well.

Cons:
- There is a relatively large body of work on self-distillation however there is no comparison against them, neither on explaining how they differ nor empirically. A non-exhaustive list:
[A] Be Your Own Teacher: Improve the Performance of Convolutional Neural Networks via Self Distillation, Zhang et al, ICCV 2019
[B] Self-Knowledge Distillation with Progressive Refinement of Targets, Kim et al, ICCV 2021
[C] MetaDistiller: Network Self-Boosting via Meta-Learned Top-Down Distillation, Liu et al, ECCV 2020
[D] Regularizing Class-Wise Predictions via Self-Knowledge Distillation, Yun et al, CVPR 2020
- How does this method work when combined with other distillation losses (such as feature matching)? KD generally can boost their performance, is the same true here?
- Little comparison with state-of-the-art on knowledge distillation and while surpassing the KD baseline is important in this context its unclear if applying such method in practice will suffice. As such the authors should offer a comparison with the latest work on KD too.
- How well does this scale to larger datasets, such as ImageNet?

Minor:
Table 2, some values are wrongly in bold (ex: KD outperforms P-KD on CIFAR-10)



**Summary Of The Paper:**

The authors propose a self-distillation approach that is derived from a series of observation made regarding how LS and KD work, linking in the two together in the process. The paper then experimentally shows the superiority of the proposed approach against KD on CIFAR-10/100 and languge datasets. Important comparisons against closely-related methods are however missing.

**Summary Of The Review:**

While the paper is interesting and the idea of performing "distillation" (pseudo) is appealing the experimental section needs to be improved.

---

> ### Author Response · Authors · 2021-11-13
> **Response to Reviewer JBRd (Part 1)**
>
> We thank you for your thoughtful feedback. The weaknesses of our paper are discussed as follows.
>
> **Q:** The authors propose a self-distillation approach that is derived from a series of observation made regarding how LS and KD work, linking in the two together in the process.
> > **A:** Our motivation is not to develop a variant of self-distillation. We connect the solution to self-distillation as it has the similar mathematical form as the soft target used in self-distillation. To make it clear, we would like to summarize and highlight the process:
> >
> >>We first propose a general framework by revisiting the conventional label smoothing method. We also justify this formulation by showing it can unify the objective of conventional LS and KD in a general way.
> >
> >
> > >Then we conduct the framework as a bi-level optimization problem and give a closed-form solution which we demonstrate to be interpretable as the soft targets used in self-distillation: leveraging the output of a model with the same capacity to regularize the training. We will clarify this in the new version.
>
> **Q**: How does this method work when combined with other distillation losses (such as feature matching)? KD generally can boost their performance, is the same true here? Little comparison with state-of-the-art on knowledge distillation and while surpassing the KD baseline is important in this context its unclear if applying such method in practice will suffice.
>
> **A:**  We propose to learn LS regularization instead of a variant of KD methods. In our case, we do not have counterpart features provided by a pre-trained Teacher to do feature matching for intermediate layers. The goal of this work is to boost the performance of conventional LS to vanilla KD. To incorporate other techniques such as feature matching can be explored in future work.

---

> > ### Author Response · Authors · 2021-11-13
> > **Response to Reviewer JBRd (Part 2)**
> >
> > **Q:** There is a relatively large body of work on self-distillation [1, 2, 3, 4], however there is no comparison against them, neither on explaining how they differ nor empirically.
> >
> > >**A:** Thank you for pointing out these references and suggesting this discussion.
> > >Among the four works, the methods in [1] and [2] are most close to our solution because we all make use of the output of models to regularize the training.
> > >>[1] which introduces an additional loss to match the soft output between different samples of the same class has already been discussed and compared in the experimental part shown in Table 2.
> > >
> > >>The recently published work [2] utilizes the model trained at epoch *i* as the teacher to regularize the training at epoch *i+1* along with annealing techniques which is quite interesting. The difference emerges as it still requires a separated model as *teacher* or saving all soft labels generated at epoch *i* for the next epoch training.
> > >
> > >>Both [3] and [4] apply regularization on the intermediate layers by using the final output as soft teacher target for all intermediate outputs which is an orthogonal direction. We will clarify the difference in the related work of the new version.
> >
> > **Reference**
> >
> > [1] Regularizing Class-Wise Predictions via Self-Knowledge Distillation. Yun et al, CVPR 2020
> >
> > [2] Self-Knowledge Distillation with Progressive Refinement of Targets, Kim et al, ICCV 2021
> >
> > [3] Be Your Own Teacher: Improve the Performance of Convolutional Neural Networks via Self Distillation, Zhang et al, ICCV 2019
> >
> > [4] MetaDistiller: Network Self-Boosting via Meta-Learned Top-Down Distillation, Liu et al, ECCV 2020

---

> ### Author Response · Authors · 2021-11-22
> **Additional experimental results**
>
> We conducted more experiments to compare the latest work PS-KD.
>
> |   |   | CIFAR100 |   |   |   | CIFAR10 |   |
> |---|---|---|---|---|---|---|---|
> |   | MobileNetv2 | ResNet18 | ResNet50 |   | MobileNetv2 | ResNet18 | ResNet50 |
> | LS | 68.69 | 77.67  | 79.16 |   | 90.82 | 95.02 | 95.10  |
> | PSKD | 70.94  | **78.36**  | **79.83**  |   | **91.56**  | 95.14  | 95.39  |
> | Pseudo-KD | **71.05**  | 78.10  | 79.76  |   | 91.53  | **95.21**  | **95.43**  |
>
> We also conduct experiments on ImageNet. Due to the time limit, we only repeated experiments 2 times.
>
> | ImageNet | ResNet18 | ResNet50 |
> |---|---|---|
> | LS | 70.11 | 76.23 |
> | PS-KD | **70.42** | **76.61** |
> | Pseudo-KD | 70.36 | 76.45
>
> Our method can give a comparable performance with neither time nor space complexity increases.

---

### Official Review · Reviewer_piYX · 2021-11-03

**Correctness:** 4
**Technical Novelty And Significance:** 3
**Empirical Novelty And Significance:** 2
**Recommendation:** 6
**Confidence:** 3

**Main Review:**

Strength:
1. The paper proposes a new formulation of an objective function for instance-specific label smoothing regularization to unify label smoothing (LS) regularization and knowledge distillation (KD), which is quite interesting.

2. It is good to find a closed-form solution for the inner level of the bi-level optimization problem of the proposed objective function.

3. Experiments have been done with data from different domains (images and texts) and have confirmed the effectiveness of the proposed objective function.

Weakness:
Presentation of the experimental results can be improved:
In Tables 4.2 and 4.4, it would be interesting to also show the standard deviation of the test accuracy/GLUE scores. The results of the propose Pseudo-KD model does not seem particularly attractive in these two tables.

Minor presentation issues:
Page 5: "in the second phase, we fix the" => "in the second stage, we fix"
Page 6: It would be good to add references for the KD-Shuffle and KD-29 models.


**Summary Of The Paper:**

This paper proposes a new formulation of an objective function for label smoothing regularization to unify label smoothing (LS) regularization and knowledge distillation (KD). The objective function learns an instance-specific label smoothing regularization while training a model towards its prediction target. When the instance-specific label smoothing regularization uses a simple uniform distribution for all training samples, the objective function falls back to traditional LS. When the instance-specific label smoothing regularization uses the output distribution of a teacher model, the objective function learns a KD model. The paper formulates the learning of the optimal instance-specific label smoothing regularization as a bi-level optimization problem and derived a closed-form solution for the inner level of the problem. Experimental results on both image classification and natural language understanding tasks show that the proposed objective and learning algorithm are effective and yield results comparable to the baseline LS and KD models.

**Summary Of The Review:**

The paper proposes a new formulation of an objective function for instance-specific label smoothing regularization to unify label smoothing (LS) regularization and knowledge distillation (KD), which is quite interesting but is also largely based on the CVPR 2020 work by Yuan et al. The paper further formulates the proposed objective function as a bi-level optimization problem and finds a closed-form solution for the inner level of the problem. Experiments confirmed the effectiveness of the proposed objective function on both image classification and natural language understanding tasks, while the improvements over the baselines are somewhat limited. Overall this is a solid study with reasonable contributions.

---

> ### Author Response · Authors · 2021-11-13
> **Response to Reviewer piYX**
>
> We thank you for your positive assessment and thoughtful comments. Besides, thank you for pointing out the presentation weakness and issues. We will fix the expression and update the table results in the revised version.
>
> We are happy to answer any other questions you may have.

---

### Official Review · Reviewer_bwuZ · 2021-11-04

**Correctness:** 3
**Technical Novelty And Significance:** 4
**Empirical Novelty And Significance:** 3
**Recommendation:** 5
**Confidence:** 4

**Main Review:**

Pros:
1. The paper is well-written and well-organized. Codes are provided.
2. The paper proposes an adaptive label smoothing method, which is novel to some extent.
3. The proposed bi-level framework and its implementation look reasonable.

Cons:

1. The motivation of the paper is my main concern. The method is called Pseudo Knowledge Distillation and the authors claim the method bridges the gap between label smoothing and knowledge distillation, however it seems to have no connection to Knowledge Distillation. The generated smoothed labels don't consider the relevance of classes which is defined as the dark knowledge in [1]. The motivation of adding a KL divergence term into Eq.7 is not elaborated clearly.
2. The effectiveness of the proposed method is not convincing. The improvement of Pseudo-KD in practice is not significant.
3. Some baselines are missing, such as Deep Mutual Learning [2] and Knowledge Refinery [3].

[1] Geoffrey E. Hinton, Oriol Vinyals, Jeffrey Dean. Distilling the Knowledge in a Neural Network
[2] Zhang, Ying, Tao Xiang, Timothy M. Hospedales, and Huchuan Lu. Deep mutual learning. CVPR2018
[3] Qianggang Ding, Sifan Wu, Tao Dai, Hao Sun, Jiadong Guo, Zhang-Hua Fu, Shutao Xia. Knowledge Refinery: Learning from Decoupled Label. AAAI 2021

**Summary Of The Paper:**

The paper proposes a novel teacher-free label regularization method named Pseudo-KD which bridges the gap between label smoothing and knowledge distillation. The proposed bilevel programming framework is implemented by an alternating two-stage process. The optimal smoothed labels are generated in the upper optimization, and the backbone network is optimized by the inner loop. The overall framework is equivalent to self-distillation without the need for extra computational cost. The authors claim that the extensive experiments on both image classification and natural language understanding tasks validate the effectiveness of the proposed method.

**Summary Of The Review:**

The idea is interesting and novel enough. However, considering the motivation and the experimental results, I vote for "5: marginally below the acceptance threshold".

---

> ### Author Response · Authors · 2021-11-13
> **Response to Reviewer bwuZ (Part 1)**
>
> We thank you for your thoughtful feedback. The weaknesses of our paper are discussed as follows.
>
> **Q:** The method is called Pseudo Knowledge Distillation while it seems to have no connection to Knowledge Distillation. The generated smoothed labels don't consider the relevance of classes which is defined as the dark knowledge in [1].
>
> > **A:** Knowledge Distillation typically leverages the output of a pre-trained teacher model to regularize the student training. The teacher predictions over non-target classes are considered to be useful which indicates the similarities between different classes [1, 2]. Namely, the relevance information of classes or dark knowledge consists in the outputs of a trained model (teacher).
> > We demonstrate our solution Eq.11 can be rewritten as Eq.14:
> >$P^*_{ls}(j\vert x_i) = \frac{\exp({\frac{z_{i,j}}{\tau}})}{\sum_{j'=1}^K \exp({\frac{z_{i,j'}}{\tau}})}$
> >
> >which applies the softmax over logits of the current model with a temperature factor and is similar to the soft targets used in KD. To summarize, our solution leverages its transformed predictions as prior knowledge over the class space. The knowledge distillation is *pseudo* because we do not require a teacher-student framework for transferring the dark knowledge.
>
> **Reference:**
>
> [1]: Geoffrey E. Hinton, Oriol Vinyals, Jeffrey Dean. Distilling the Knowledge in a Neural Network
>
> [2]: Rafael Muller, Simon Kornblith, and Geoffrey E. Hinton. When Does Label Smoothing Help?

---

> > ### Author Response · Authors · 2021-11-13
> > **Response to Reviewer bwuZ (Part 2)**
> >
> > **Q:** The effectiveness of the proposed method is not convincing. The improvement of Pseudo-KD in practice is not significant.
> >
> > **A:**  We repeat 3 more times experiments on CIFAR100 with MobileNetv2, ResNet18 and ResNet50 for LS and Pseudo-KD, then conduct Wilcoxon signed-rank test suggested by [1, 2].
> >
> > |  LS vs Pseudo-KD| p-value |
> > |---|---|
> > | MobileNetv2 | 0.008 |
> > | ResNet18 | 0.023 |
> > | ResNet50 | 0.016 |
> >
> > The p-values are consistently less than 0.05. Regarding the limited improvement in terms of accuracy, although it seems not significant in value, it is achieved in a simple online manner without extra time or space complexity.
> >
> > **Reference:**
> >
> > [1] Demsar, J. Statistical comparisons of classifiers over multiple data sets. Journal of Machine learning research, 7 (Jan):1–30, 2006.
> >
> > [2] Xavier Bouthillier et al. Accounting for Variance in Machine Learning Benchmarks.

---

> > > ### Author Response · Authors · 2021-11-22
> > > **Additional Experimental Results**
> > >
> > > **Q:** Some baselines are missing, such as Deep Mutual Learning [2] and Knowledge Refinery [3]
> > >
> > > *A:* Thank you for pointing out these references and suggesting this comparison.
> > > [2] propose to train an ensemble of students without a static teacher, which benefits from learning and teaching among different students collaboratively while we formulate a general LS framework and derive a new instance-specific LS method. We believe this is an orthogonal direction as our goal is to boost the performance of LS instead of KD.
> > > [3] is a recently published work which proposes a method *KR-LS* to capture the relation of classes by introducing decoupled labels table which increases $O(K\times K)$ space complexity.
> > >
> > > |   |   | CIFAR100 |   |   |   | CIFAR10 |   |
> > > |---|---|---|---|---|---|---|---|
> > > |   | MobileNetv2 | ResNet18 | ResNet50 |   | MobileNetv2 | ResNet18 | ResNet50 |
> > > | LS | 68.69 | 77.67  | 79.16 |   | 90.82 | 95.02 | 95.10  |
> > > | KR-LS | 70.25  | 77.82  | 79.48  |   | 90.67  | 94.76  | 95.30  |
> > > | Pseudo-KD | 71.05  | 78.10  | 79.76  |   | 91.53  | 95.21  | 95.43  |

---

### Official Review · Reviewer_CUiW · 2021-11-07

**Correctness:** 3
**Technical Novelty And Significance:** 2
**Empirical Novelty And Significance:** 2
**Recommendation:** 5
**Confidence:** 3

**Main Review:**

This paper presents an online self-distillation method. The paper is well-written and I enjoy reading it. However, the proposed link between label smoothing and knowledge distillation, in my opinion, is still based on previous research findings (e.g., Yuan et al., 2020). When compared to previous research, it would be interesting to see what new insights can be gained from the newly proposed loss function (7) (other than adding the smoothing kl term to the original ce loss). Or, to put it another way, I believe it is nearly equivalent if we use the formulation proposed in previous work (Yuan et al., 2020). However, in section 3.2, this paper presents a closed-form solution to the two-stage optimization problem.

In the evaluation part, it would be great if authors could compare Pseudo-KD to the recent efforts on self-distillation (instance-specific label smoothing), for example:
   Zhang, Zhilu, and Mert R. Sabuncu. "Self-distillation as instance-specific label smoothing."
   Yuan, Li, et al. "Revisiting knowledge distillation via label smoothing regularization."









**Summary Of The Paper:**

This paper presents an instance-specific label smoothing (self-distillation) technique, Pseudo-KD, formulated as a two-stage optimization problem. On popular CV and NLU benchmarks, authors conducted experiments to compare Pseudo-KD against LS, KD, and various teacher-free methods, demonstrating the usefulness and efficiency of the proposed technique.

**Summary Of The Review:**

1. novelty is limited compared to the recent work on self distillation and instance-specific label smoothing
2. missing relevant baselines in the evaluation section

---

> ### Author Response · Authors · 2021-11-13
> **Response to Reviewer CUiW (Part 1)**
>
> We thank you for your thoughtful feedback. The weaknesses of our paper are discussed as follows.
>
> **Q1.** Novelty is limited compared to the recent work on self distillation and instance-specific label smoothing. The proposed link between label smoothing and knowledge distillation, in my opinion, is still based on previous research findings (e.g., Yuan et al., 2020).
>
> **A:** Our work is inspired by recent findings [yuan2020revisiting,  Zhang & Sabuncu, 2020] but not limited to revealing the connection between Label Smoothing (LS) and Knowledge Distillation (KD) again. Specifically, we summarize the difference in three-fold:
>
> **Motivation.** We generalize the smoothing distribution to a general form $P_{ls}$. This distribution is intractable and arbitrary, so we choose a Uniform distribution to constrain $P_{ls}$ in the KL term as we would like $P_{ls}$ to contain more information on non-target classes in Eq 8.
>
> $\frac{1}{N}\sum_{i=1}^N   E_{\hat{P}_{ls}} [-\log p_\theta(j|x_i)]   +  \beta KL (P _{ls} (\cdot | x_i) || U( \cdot ))$
>
>  Actually, other distributions can be explored in the future to replace the uniform distribution in the KL term for the purpose of introducing other inductive biases over $P_{ls}$.
>
> **Post-hoc interpretability.** To justify our framework, we show two special cases by replacing $P_{ls}$ to $U$ and $P_T$ respectively which leads to the conventional LS and KD objectives. This formulation not only confirms the relationship of LS and KD from a different perspective but also justifies that our unified framework is reasonable.
>
> **Solution.** Based on our framework, we further propose to learn the smoothing distribution along with training by formulating it as a bilevel optimization problem and we derive a deterministic and interpretable solution Eq.11 for the first stage. Besides, we also give an online implementation and show its effectiveness and efficiency.

---

> ### Author Response · Authors · 2021-11-13
> **Response to Reviewer CUiW (Part 2)**
>
> **Q2:** It would be great to compare Pseudo-KD to the recent efforts on self-distillation (instance-specific label smoothing), for example, Zhang, Zhilu, and Mert R. Sabuncu. "Self-distillation as instance-specific label smoothing." Yuan, Li, et al. "Revisiting knowledge distillation via label smoothing regularization."
>
> **A**: We thank the reviewer for pointing this out. We already show comparisons with these works in Table 2. To make it clear, Beta-LS and Tf-reg refer to methods proposed in these works, respectively. Our method has better performance than these methods.

---

### Author Response · Authors · 2021-11-23
**General Response**

We thank all of the reviewers' comprehensive remarks and helpful feedback. To address the concerns from reviewers, we have made the following main changes in our draft:
- We revise the related work to include references mentioned by reviewers. Some of these works have previously been discussed and compared in previous versions (Beta-LS [1] and CS-KD [2]). While two works (KR-LS[3] and PS-KD[4]) are recently published and one of them hasn't published their code officially (KR-LS). The other approaches, which apply regularization on intermediate layers or collectively train an ensemble of models to improve performance, are orthogonal to learning Label Smoothing Regularization.

- We modify and highlight the motivation for our new formulation and refine the remark to justify our framework by showing the relationship of our objective with LS and KD.

- We add the standard deviations of all our previous experiments in Table 2. We also empirically compare our method with two more recently published works KR-LS and PS-KD for three different methods on CIFAR100 and CIFAR10.

Thanks again for all the valuable comments and suggestions.

**Reference**

[1] Zhang et al, Self-distillation as instance-specific label smoothing. NeurIPS 2020.
[2] Yun et al, Regularizing Class-Wise Predictions via Self-Knowledge Distillation.  CVPR 2020.
[3] Ding et al, Knowledge Refinery: Learning from Decoupled Label. AAAI 2021.
[4] Kim et al, Self-Knowledge Distillation with Progressive Refinement of Targets, ICCV 2021.

---

### Decision · Program_Chairs · 2022-01-20

**Decision:**

Reject

**Comment:**

This work proposes an instance-specific label smoothing method, which is formulated as a two-stage optimization problem for finding the optimal label smoothing. The authors show that the proposed approach can be equivalent to an efficient variant of self-distillation techniques (i.e. no need to store the parameters or the output of a trained model). Experiments on image classification (CIFAR-10 and CIFAR-100) and natural language understanding datasets (the GLUE benchmark) demonstrate that our method is competitive against strong baselines.

The reviewers find the proposed approach reasonable, and the presentation clear. However, they all rated the paper as borderline, due to some concerns that the submission has in its current form. These include limited novelty (by CUiW) [the link between label smoothing and knowledge distillation is largely based on previous research findings (e.g., Yuan et al., 2020)], nonconvincing results on the effectiveness of the proposed method (by bwuZ) [Improvement of Pseudo-KD in practice is not significant in terms of test accuracy gains], and lack of comparison with some recent related methods (by JBRd as well as other reviewers). The authors responded to these (and other concerns), but this did not convince the reviewers about the concerns listed above. I recommend the authors to resubmit after addressing these issues.